# Usefulness of 18f-FDG PET-CT in Staging, Restaging, and Response Assessment in Pediatric Rhabdomyosarcoma

**DOI:** 10.3390/diagnostics10121112

**Published:** 2020-12-21

**Authors:** Davide Donner, Paola Feraco, Linda Meneghello, Barbara Rombi, Lorena Picori, Franca Chierichetti

**Affiliations:** 1Nuclear Medicine Unit, Santa Chiara Hospital, Azienda Provinciale per i Servizi Sanitari della Provincia Autonoma di Trento, 38123 Trento, Italy; lorena.picori@apss.tn.it (L.P.); franca.chierichetti@apss.tn.it (F.C.); 2Unit of Neuroradiology, Santa Chiara Hospital, Azienda Provinciale per i Servizi Sanitari della Provincia Autonoma di Trento, 38123 Trento, Italy; paola.feraco@apss.tn.it; 3Department of Experimental, Diagnostic and Specialty Medicine (DIMES), University of Bologna, 40126 Bologna, Italy; 4Department of Pediatrics, Santa Chiara Hospital, Azienda Provinciale per i Servizi Sanitari della Provincia Autonoma di Trento, 38123 Trento, Italy; linda.meneghello@apss.tn.it; 5Proton Therapy Center, Azienda Provinciale per i Servizi Sanitari della Provincia Autonoma di Trento, 38123 Trento, Italy; barbara.rombi@apss.tn.it

**Keywords:** ^18^F-FDG PET-CT, rhabdomyosarcoma, pediatric, staging, restaging, therapy assessment, follow-up

## Abstract

Rhabdomyosarcoma is the most common soft-tissue sarcoma of childhood. Despite clinical advances, subsets of these patients continue to suffer high morbidity and mortality rates associated with their disease. Following the European guidelines for ^18^F-FDG PET and PET-CT imaging in pediatric oncology, the routine use of ^18^F-FDG PET-CT may be useful for patients affected by rhabdomyosarcoma, in staging, in the evaluation of response to therapy, and for restaging/detection of relapse. The European Pediatric Protocols are very old, and for staging and restaging, they recommend only radionuclide bone scan. The ^18^F-FDG PET-CT exam is listed as an optional investigation prescribed according to local availability and local protocols in the investigations panel required at the end of the treatment. We present two cases highlighting the usefulness of ^18^F-FDG PET-CT in managing pediatric patients affected by rhabdomyosarcoma, providing some bibliographic references.

**Figure 1 diagnostics-10-01112-f001:**
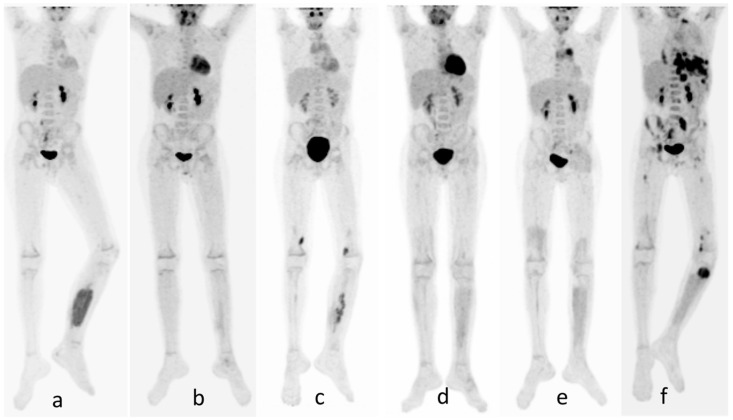
Patient 1. A 9-year-old boy with alveolar rhabdomyosarcoma [1,2,3,4,5,6,7] of the extremities (left popliteal region) was diagnosed by MRI and by biopsy (negative sentinel inguinal lymph node), and a staging 18F-FDG PET-CT [8,9] was performed. The whole-body PET-CT exam (**a**) showed a pathologic uptake (SUVmax 5.1 with Lesion SUVmax/Liver SUVmax Tumor Liver Ratio (TLR) <4.6) at the middle third of the left leg in the deep posterior compartment, involving the long flexor muscle of the fingers and the long flexor muscle of the big toe (PET-CT and MRI fusion imaging) and absence of other foci of pathological uptake in the remaining explored field of view. Brain MRI was negative for secondary lesions. Chemotherapy, according to EpSSg RMS 2005 protocol High-risk group G, was performed and for local control, photon radiotherapy (50.4 Gy in 28 fractions in the left calf). There was no surgery because it was complete remission at the control after the third cycle of chemotherapy. A month after the end of chemotherapy, to assess the response, an MRI of the calf and a whole-body 18F-FDG PET-CT were repeated [9]. MRI showed an area of mild enhancement (9 × 20 mm) at the long flexor muscle of the big toe, which was interpreted as a post-treatment tissue reaction with prescription of a short follow-up, and PET-CT (**b**) findings were consistent with a complete metabolic response. At the follow-up (control) MRI, the calf’s finding was unchanged, and therefore a biopsy was performed, which showed recurrence of a known disease. A restaging 18F-FDG PET-CT (**c**) was done [7,10], which documented the appearance of some areas of mild FDG uptake in the soft tissues of the distal third of the left leg (SUVmax 4.2), in the left gluteus maximum muscle of a hypermetabolic lymph node in the ipsilateral popliteal fossa, and another lymph node with moderate FDG uptake (SUVmax 5.89; short-axis diameter 20 mm) near the distal head of the right (contralateral) medial vastus muscle. The calf disease relapse and the progression of disease to omo (left) and, totally unexpected, contralateral (right) leg’s lymph nodes and to left gluteus maximus muscle, documented by 18F-FDG PET-CT, were confirmed by MRI and biopsy. A second therapeutic line was decided with 8 IrVAC cycles (Irinotecan, Ciclofosfamide, Vincristine, and Actinomycin D), plus another two cycles with Vincristine and Irinotecan with concomitant proton radiotherapy irradiation of all disease sites, respectively: 54 Gy, in 30 fractions, in the left calf and 50.4 Gy, in 28 fractions, to the left popliteal region, the left gluteal muscle, and the right medial vastus muscle. An early re-assessment of the treated sites of disease was performed using PET-CT [10] (**d**) and MRI. Both exams were consistent with disease remission. Despite the good results of recent radio/chemotherapy, and considering the previous diffuse relapse disease, low-dose maintenance chemotherapy with VP16 (Eposide) was started. Three months later, a further PET-CT (**e**) showed a metabolic progression at the thoracic level, and six months after this examination (**f**), the PET-CT was consistent with a progressive disease with bone involvement, lymph nodes, and lungs metastasis. The patient died 25 months after the initial diagnosis.

**Figure 2 diagnostics-10-01112-f002:**
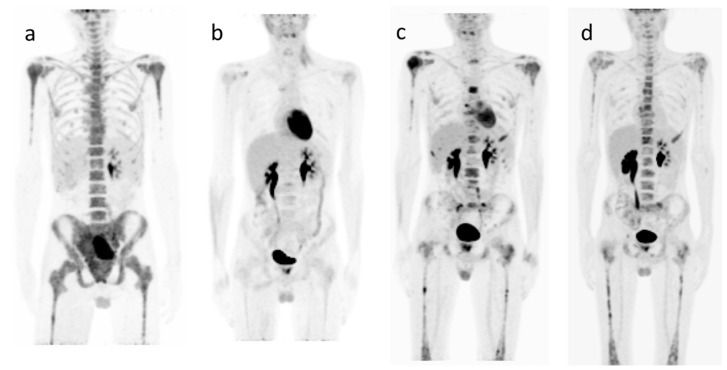
Patient 2. An 18-year-old boy entered the pediatric emergency room with abdominal pain and occlusive symptoms. Contrast-enhanced CT documented bilateral sub-cent metric lung micro-nodules and a lesion with maximum diameters of 15 × 8 × 10 cm in the right median-paramedian pelvic region, surrounding the right arteriovenous iliac axis and the ipsilateral ureter with associated hydroureteronephrosis. Concomitant lymphadenopathies were observed along the left iliac axis, in the external iliac region of the same side, and at the ipsilateral inguinal level. Areas of cortical-subcortical bone-rarefaction suspected of metastasis in the sternum, shoulder blades, femoral necks, and wedge deformation of the 9th thoracic vertebra were reported as well. (**a**) A staging ^18^F-FDG PET-CT [8,9] was therefore performed. Maximum intensity projection showed a high glucose uptake both to the known right pelvic mass (SUVmax 13.5 and SUVmax lesion/SUVmax liver = Tumor Liver Ratio (TLR) >4.6), to the level of common and right internal and external iliac lymph nodes, and to other lymph nodes of the obturator region of the same side. Finally, the scan documented an intense bone marrow uptake that involved multiple sites, especially spine and pelvis, but also upper and lower limbs and a diffuse uptake to testes. As collateral findings, increased FDG uptake was evident to the 9th thoracic vertebra and to the right kidney due, respectively, to recent neurosurgical intervention (diagnostic laminectomy) and to functional renal failure treated with double J catheter placement. PET-CT results, i.e., a highly metabolically active primary tumor with a TLR greater than 4.6, together with the presence of metabolically active lymph nodes and the involvement of distant sites, were indicative of significantly lower survival rates and were consistent with an aggressive metastatic oncological disease [11,12,13]. The specimen (laminectomy) showed an alveolar rhabdomyosarcoma (ARMS) with chromosomal translocations t(2;13) (q35;q14), which results in the expression of an oncogenic fusion protein PAX3 transcription factor with the transcriptional activation domain of FOXO1, which is more aggressive, prone to metastasis, and carries a poorer prognosis compared to the more common embryonal rhabdomyosarcoma subtype [11,12,14]. The staging ^18^F-FDG PET-CT confirmed the aggressiveness of the tumor (due to primary tumor high metabolic intensity) [13,15,16] and the metastatic spread to lymph nodes and bones and, with TLR, contributed to predicting the disease outcome [13,15,16]. (**b**) A second ^18^F-FDG PET-CT exam was performed during the last (6/6) chemotherapy cycle (Ir-IVA [Irinotecan]-[Ifosfamide, Vincristine, and dactinomycin]) to assess the response to the treatment 11. The PET-CT proved a partial response, showing a significant reduction in volume and metabolic activity of the retroperitoneal pelvic primary tumor (SUVmax 3 vs. SUVmax 13.5 and a Delta volume reduction >60%), a persistence of mild to discrete pathological metabolic activity (Lesion SUVmax < Liver SUVmax) in the bone marrow especially at sternum level, and complete metabolic response with a morphologic normalization regarding lymph nodes, but a pathological prostate uptake was also reported. Photon radiotherapy at the pelvic and sternum level, followed by maintenance chemotherapy, was performed. (**c**) A third ^18^F-FDG PET-CT scan was performed one month after the end of photon radiotherapy and during maintenance chemotherapy with cyclophosphamide and vinorelbine for re-staging [7] and assessment of therapies response [10,17,18]. The exam showed a partial response at the sternum and primary tumor site, but skeleton disease relapse/progression. (**d**) Three months after the last nuclear medical investigation, following the suspicion of brain progression, confirmed by the contrast-enhanced CT (metastatic leptomeningeal disease), a new PET-CT was performed for a complete restaging. This examination revealed persistent active disease at the retroperitoneal pelvic level and a further skeletal progression. The patient died 18 months after the initial diagnosis.

In the first one (Figure 1) of the two presented cases, the unexpected lymph-nodes metastatic involvement of the contralateral leg and the evidence of distance metastasis to the left gluteus maximus muscle, documented by ^18^F-FDG PET-CT, with the data of the evolution of the disease provided by PET exams, are the significant contributions to the management of the patient. In the second case (Figure 2), the ^18^F-FDG PET-CT played a major role in giving both an accurate evaluation of the disease extent (staging and restaging) and a metabolic prognostic factor TLR that Baum et al. [14] have correlated with the survival rate [19]. Our two cases are not a suggestion to use only serial ^18^F-FDG PET-CT in the follow-up of these patients, but they represent an invitation to consider, as many works suggest [20,21,22,23,24], that the provided metabolic information could be extremely useful in the disease management, with an estimated 21% overall potential benefit of PET-CT over CT/MRI, in the “upstaging” of high-grade disease, as Macpherson et al. demonstrated in a recent work [20].

The radiation exposure is hugely restrained, since the CT automated exposure control pediatric acquisition protocols [25] and the optimization of ^18^F-FDG pediatric dose protocols [26] are routinely used. The two described cases are examples of how PET-CT can play a strategic role for RMS pediatric patients and how, in view of evidence in the literature, it is increasingly necessary for this examination to be included in the diagnostic rhabdomyosarcoma protocols.

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
