# Peer review of "Usefulness of 18f-FDG PET-CT in Staging, Restaging, and Response Assessment in Pediatric Rhabdomyosarcoma"

_diagnostics, 2020, doi:10.3390/diagnostics10121112_

Round 1

Reviewer 1 Report

Donner et al present a series of two cases of pediatric rhabdomyosarcoma followed by 18F-FDG PET/CT. The main finding of this study is that PET/CT can be a useful tool for the metabolic assessment of this tumor entity, with potential implications in the staging, re-staging, and treatment response assessment.

The results of this study are interesting but not novel, since previous papers have highlighted the value of the modality in rhabdomyosarcoma.

The following issues constitute weaknesses of the study:

  • The term CT-PET is new to me. I have never seen it before in the literature. I strongly suggest using the ‘conventional’ term PET-CT.
  • The Abstract is weak, serving actually as a mere introduction, while the findings of the study are confined in one and a half line (ln 34-35). Moreover, it is unheard to cite articles in a paper abstract. I am not aware if this is a common practice in this article type of the journal (Interesting Images). If not, this should be modified.
  • In general, the text is not well written, and is thus difficult to follow. The authors provide a detailed description of the PET scans’ findings, without any substantial discussion of the existing literature in the field.
  • Figure 2: Please provide also the respective transversal PET/CT images in the follow-up scans, similar to baseline imaging (2a).
  • Ln 71-72: It seems to me that in Figure 1d, there is clear disease remission. Why do the authors mention ‘persistence of disease’? If persistence is indeed the case, then please provide more detailed/better images.
  • Ln 72-73. It is difficult to understand what the authors mean with the provided SUV values in brackets.
  • Ln 75-76: The authors mention that the metabolic findings were unchanged in Fig. 1e. However, we can clearly see a, newly emerging, intensive FDG accumulation at the thoracic level. This is most likely a metabolic disease progression.
  • Ln 96: please explain D9. Do the authors mean the 9th thoracic vertebra?
  • Ln 109: When during chemotherapy, i.e. after how many cycles? Was this considered an interim PET/CT?
  • To show the potential added value of 18F-FDG PET/CT in response assessment it must be compared to the standard of care, i.e. MRI in most centres. Such comparison data, incl. the respective Figures, showing that PET/CT is as good or even better than MRI would strengthen the message of the article.
  • Based on these two cases, do the authors suggest the inclusion of serial PET/CT scanning in the management of these patients? This point is of significance considering the additional radiation exposure of the pediatric patients and the increased cost associated with the examination(s). These aspects should be taken into consideration.

Author Response

I thank a lot the reviewers, especially the first, for the corrections and advice provided.

I have entered all the corrections/indications reported. I added some considerations in the final stage of the work. I have added 5 new references, 3 of which are very recent, as requested by all three reviewers.

Finally, I decided to simplify the images provided in case 2, trying to underline the role of PET alone, through MIP images.

All the corrections suggested and the considerations added in the final part of the work have been highlighted in red to make them better evaluable.

Donner et al present a series of two cases of pediatric rhabdomyosarcoma followed by 18F-FDG PET/CT. The main finding of this study is that PET/CT can be a useful tool for the metabolic assessment of this tumor entity, with potential implications in the staging, re-staging, and treatment response assessment.

The results of this study are interesting but not novel, since previous papers have highlighted the value of the modality in rhabdomyosarcoma.

The following issues constitute weaknesses of the study:

  • The term CT-PET is new to me. I have never seen it before in the literature. I strongly suggest using the ‘conventional’ term PET-CT. I did it (see attached file)
  • The Abstract is weak, serving actually as a mere introduction, while the findings of the study are confined in one and a half line (ln 34-35). Moreover, it is unheard of to cite articles in a paper abstract. I am not aware if this is a common practice in this article type of journal (Interesting Images). If not, this should be modified. I fixed it (see attached file)
  • In general, the text is not well written and is thus difficult to follow. The authors provide a detailed description of the PET scans’ findings, without any substantial discussion of the existing literature in the field. I added some discussion at the end of the paper (see attached file)
  • Figure 2: Please provide also the respective transversal PET/CT images in the follow-up scans, similar to baseline imaging (2a). I choose to simplify providing the MIP PET images only (see attached file)
  • Ln 71-72: It seems to me that in Figure 1d, there is clear disease remission. Why do the authors mention ‘persistence of disease’? If persistence is indeed the case, then please provide more detailed/better images. I apologize for the mistake. I fixed it (see attached file).
  • Ln 72-73. It is difficult to understand what the authors mean with the provided SUV values in brackets. I erased the text (see attached file).
  • Ln 75-76: The authors mention that the metabolic findings were unchanged in Fig. 1e. However, we can clearly see a, newly emerging, intensive FDG accumulation at the thoracic level. This is most likely a metabolic disease progression. Sorry, I fixed it, (see attached file).
  • Ln 96: please explain D9. Do the authors mean the 9th thoracic vertebra? The text has been modified (see attached file).
  • Ln 109: When during chemotherapy, i.e. after how many cycles? Was this considered an interim PET/CT? during the last (6/6) chemotherapy cycle (see attached file).
  • To show the potential added value of 18F-FDG PET/CT in response assessment it must be compared to the standard of care, i.e. MRI in most centers. Such comparison data, incl. the respective Figures, showing that PET/CT is as good or even better than MRI would strengthen the message of the article. I added some considerations and I quoted a paper to try to strengthen the message of the article. Lines 107-117 (see attached file).
  • Based on these two cases, do the authors suggest the inclusion of serial PET/CT scanning in the management of these patients? This point is of significance considering the additional radiation exposure of the pediatric patients and the increased cost associated with the examination(s). These aspects should be taken into consideration. I added some considerations. Lines 107-117 (see attached file). 

Reviewer 2 Report

The authors describe 2 cases that highlight the usefulness of 18F-FDG 34 CT-PET in the management of paediatric patients affected by rhabdomyosarcoma. The cases clearly describe clinical practice and are illustrative. However, the only reference to current guidelines is a reference to the RMS2005 protocol, that has an outdated imaging guideline. The manuscript migh benefit from a systematic review of the literature on the diagnostic and prognostic value of FDG-PET (response) in rhabdomyosarcoma treatment.

Author Response

The authors describe 2 cases that highlight the usefulness of 18F-FDG 34 CT-PET in the management of paediatric patients affected by rhabdomyosarcoma. The cases clearly describe clinical practice and are illustrative. However, the only reference to current guidelines is a reference to the RMS2005 protocol, which has an outdated imaging guideline. The manuscript might benefit from a systematic review of the literature on the diagnostic and prognostic value of FDG-PET (response) in rhabdomyosarcoma treatment.

I thank a lot the reviewers for the corrections and advice provided. I have entered all the corrections/indications reported. I have added 5 new references, 3 of which are very recent, as requested by all three reviewers.

Finally, I decided to simplify the images provided in case 2, trying to underline the role of PET alone, through MIP images.

All the corrections suggested and the considerations added in the final part of the work have been highlighted in red to make them better evaluable.

I added 5 references 3 of which highlighted that the RMS2005 protocol has an outdated imaging guideline.

  1. E. Macpherson, S. Pratap, H. Tyrrell, M. Khonsari, S. Wilson, M. Gibbons, D. Whitwell, H. Giele, P. Critchley, L. Cogswell, S. Trent, N. Athanasou, K.M. Bradley, A.B. Hassan. Retrospective audit of 957 consecutive 18F-FDG PET-CT scans compared to CT and MRI in 493 patients with different histological subtypes of bone and soft tissue sarcoma. Clin Sarcoma Res. 2018 Apr 9;8:9. doi: 10.1186/s13569-018-0095-9. PMID: 30116519; PMCID: PMC6086048.
  2. Annovazzi, S. Rea, C. Zoccali, R. Sciuto, J. Baldi, V. Anelli, M. G. Petrongari, E. Pescarmona , R. Biagini and V. Ferraresi. Diagnostic and Clinical Impact of 18F-FDG PET/CT in Staging and Restaging Soft-Tissue Sarcomas of the Extremities and Trunk: Mono-Institutional Retrospective Study of a Sarcoma Referral Center. J. Clin. Med. 2020, 9, 2549; doi:10.3390/jcm9082549
  3. Vlenterie, W. J. Oyen, N. Steeghs, I. M. E. Desar, R. B. Verheijen, A. M. Koenen, W. Grootjans, L. F. De Geus-Oei, N. P. Van Erp, W. T. Van Der Graaf. Early Metabolic Response as a Predictor of Treatment Outcome in Patients With Metastatic Soft Tissue Sarcomas. Anticancer Res March 2019, 39 (3), 1309-1316; doi: 10.21873/anticanres.13243
  4. Elmanzalawy, R. Vali, G. B. Chavhan, A. A. Gupta, Y. Omarkhail, A. Amirabadi, A. Shammas. The impact of 18F-FDG PET on initial staging and therapy planning of pediatric soft-tissue sarcoma patients. Pediatr. Radiol. 2020;50:252–260. doi: 10.1007/s00247-019-04530-1.
  5. Bertolini, A. Palmieri, M. C. Bassi, M. Bertolini, V. Trojani, V. Piccagli, F. Fioroni, S. Cavuto, M. Guberti, A. Versari, S. Cola. CT protocol optimisation in PET/CT: a systematic review. EJNMMI Phys 7, 17 (2020). https://doi.org/10.1186/s40658-020-00287-x
  6. Michael Lassmann, S. Ted Treves and the EANM/SNMMI Paediatric Dosage Harmonization Working Group. Paediatric radiopharmaceutical administration: harmonization of the 2007 EANM paediatric dosage card (version 1.5.2008) and the 2010 North American consensus guidelines. Eur J Nucl Med Mol Imaging. 2014; DOI 10.1007/s00259-014-2731-9

Reviewer 3 Report

Donner at al. chose a nice topic for this interesting image. 

I suggest to the authors to improve the description of the previous literature on this topic in the discussion section.

Author Response

I suggest to the authors to improve the description of the previous literature on this topic in the discussion section.

I thank a lot the reviewers for the corrections and advice provided. I have added 5 new references, 3 of which are very recent, as requested by all three reviewers. All the corrections suggested and the considerations added in the final part of the work have been highlighted in red to make them better evaluable (see attached file).

  1. E. Macpherson, S. Pratap, H. Tyrrell, M. Khonsari, S. Wilson, M. Gibbons, D. Whitwell, H. Giele, P. Critchley, L. Cogswell, S. Trent, N. Athanasou, K.M. Bradley, A.B. Hassan. Retrospective audit of 957 consecutive 18F-FDG PET-CT scans compared to CT and MRI in 493 patients with different histological subtypes of bone and soft tissue sarcoma. Clin Sarcoma Res. 2018 Apr 9;8:9. doi: 10.1186/s13569-018-0095-9. PMID: 30116519; PMCID: PMC6086048.
  2. Annovazzi, S. Rea, C. Zoccali, R. Sciuto, J. Baldi, V. Anelli, M. G. Petrongari, E. Pescarmona , R. Biagini and V. Ferraresi. Diagnostic and Clinical Impact of 18F-FDG PET/CT in Staging and Restaging Soft-Tissue Sarcomas of the Extremities and Trunk: Mono-Institutional Retrospective Study of a Sarcoma Referral Center. J. Clin. Med. 2020, 9, 2549; doi:10.3390/jcm9082549
  3. Vlenterie, W. J. Oyen, N. Steeghs, I. M. E. Desar, R. B. Verheijen, A. M. Koenen, W. Grootjans, L. F. De Geus-Oei, N. P. Van Erp, W. T. Van Der Graaf. Early Metabolic Response as a Predictor of Treatment Outcome in Patients With Metastatic Soft Tissue Sarcomas. Anticancer Res March 2019, 39 (3), 1309-1316; doi: 10.21873/anticanres.13243
  4. Elmanzalawy, R. Vali, G. B. Chavhan, A. A. Gupta, Y. Omarkhail, A. Amirabadi, A. Shammas. The impact of 18F-FDG PET on initial staging and therapy planning of pediatric soft-tissue sarcoma patients. Pediatr. Radiol. 2020;50:252–260. doi: 10.1007/s00247-019-04530-1.
  5. Bertolini, A. Palmieri, M. C. Bassi, M. Bertolini, V. Trojani, V. Piccagli, F. Fioroni, S. Cavuto, M. Guberti, A. Versari, S. Cola. CT protocol optimisation in PET/CT: a systematic review. EJNMMI Phys 7, 17 (2020). https://doi.org/10.1186/s40658-020-00287-x
  6. Michael Lassmann, S. Ted Treves and the EANM/SNMMI Paediatric Dosage Harmonization Working Group. Paediatric radiopharmaceutical administration: harmonization of the 2007 EANM paediatric dosage card (version 1.5.2008) and the 2010 North American consensus guidelines. Eur J Nucl Med Mol Imaging. 2014; DOI 10.1007/s00259-014-2731-9

Round 2

Reviewer 1 Report

The author has tried to reply to most of the comments. Some of them have been addressed successfully and the manuscript has been improved.

However, there are still some unaddressed issues:

  • Abstract: I am still not comfortable with the use of all these references in the abstract section. Moreover, the term CT-PET is still used. Further, a conclusion/take-home message is lacking.
  • Unfortunately, I have to respectfully insist that the paper is not well written. I believe it would benefit from an extensive editing of the English language and style throughout the text. For example in lines 115 – 121, this single (!) sentence is extremely difficult to follow.

Author Response

Dear Reviewer about:

1) Extensive editing of English language and style required.

I modified all the text and the sentence underlined by You, and I choose to submit the manuscript to extensive editing of the English language and style provided by the Editor

2) Are the conclusions supported by the results?

I added 2 more reviews to support the conclusions, but it is difficult to talk about results with two cases. I ask the Reviewer if she/he can suggest some reviews or articles that can better highlight the use of 18FDG PET CT in RMS pediatric patients.

Best wishes,

Davide Donner

Round 3

Reviewer 1 Report

The manuscript has been improved compared to the first version. Nevertheless, there are still some issues.

Throughout the text small syntactic errors remain (e.g. ln 28 ‘the routinely use’ should be ‘ the routine use’, further mistakes in ln 75-77, 82-84 etc.). I strongly recommend the authors to perform a thorough search in the draft and proceed to the required corrections.

Moreover, some mistakes are repeated by the authors despite the previous comments. For example, I do not understand why the term CT-PET keeps being used in the abstract, while it has changed to the correct term PET-CT throughout the text.

Further, the authors keep citing literature in the abstract. I am not aware if this is a common practice in this article type of the journal (Interesting Images). Could the Editor please enlighten me on this topic? If yes, then this is no problem. If no, then this must be corrected.

Finally, the manuscript lacks a clear conclusion-paragraph. In the last two paragraphs of the article, the authors mention the contribution of PET/CT in the herein presented cases. However, this is not performed in a rational and reader-friendly manner. In other words, the conclusions are correct but not well-presented.

If these mistakes are corrected, then the work could be considered for publication.